# Computing Happiness from Textual Data

**Emad Mohamed** [1,†] and **Sayed A. Mostafa** [2,*,†]

1   Research Group in Computational Linguistics, University of Wolverhampton, Wolverhampton WV1 1LY, UK
2   Department of Mathematics & Statistics, North Carolina A&T State University, Greensboro, NC 27411, USA
*   Correspondence: sabdelmegeed@ncat.edu; Tel.: +1-336-285-3104
†   These authors contributed equally to this work.

**Abstract:** In this paper, we use a corpus of about 100,000 happy moments written by people of different genders, marital statuses, parenthood statuses, and ages to explore the following questions: Are there differences between men and women, married and unmarried individuals, parents and non-parents, and people of different age groups in terms of their causes of happiness and how they express happiness? Can gender, marital status, parenthood status and/or age be predicted from textual data expressing happiness? The first question is tackled in two steps: first, we transform the happy moments into a set of topics, lemmas, part of speech sequences, and dependency relations; then, we use each set as predictors in multi-variable binary and multinomial logistic regressions to rank these predictors in terms of their influence on each outcome variable (gender, marital status, parenthood status and age). For the prediction task, we use character, lexical, grammatical, semantic, and syntactic features in a machine learning document classification approach. The classification algorithms used include logistic regression, gradient boosting, and fastText. Our results show that textual data expressing moments of happiness can be quite beneficial in understanding the "causes of happiness" for different social groups, and that social characteristics like gender, marital status, parenthood status, and, to some extent age, can be successfully predicted form such textual data. This research aims to bring together elements from philosophy and psychology to be examined by computational corpus linguistics methods in a way that promotes the use of Natural Language Processing for the Humanities.

**Keywords:** fastText; gradient boosting; happiness; lemmatization; lexical analysis; logistic regression; parsing; topic modeling

## 1. Introduction

In the psychological sense, happiness is a state of mind that can be typically defined in terms of life satisfaction, pleasure, or positive emotional conditions. Happiness can also be seen in the sense of well-being. In this sense, hedonists define happiness as the experience of pleasure and desire theorists define happiness in terms of obtaining one's desires. Objective List theorists, unlike the previous two schools, view happiness as an objective, rather than subjective topic, and claim that some things bring us benefit regardless of our attitude towards them. For Aristotelians, who subscribe to this school, a "passive but contented couch potato may be getting what he wants, and he may enjoy it ... but he would not count as doing well, or leading a happy life" Haybron [1].

While happiness research is in the domains of psychology and philosophy, we also believe that happiness can be studied quantitatively, and we do so utilizing two fields of inquiry: computational linguistics and statistics. Our research seeks to answer some questions concerning happiness using linguistic data. The following questions guide our exploration of the *HappyDB* data which is introduced in the next section: (i) What makes people happy? (ii) Do men/women, married/unmarried people,

parents/non-parents, and people in different age groups differ in their causes of happiness? (iii) Are there linguistic differences between men and women, the married and the unmarried, parents and non-parents, the old and the young in their expression of happiness? (iv) Can we predict gender, marital status, parenthood status, and age from textual data expressing happiness?

The HappyDB data was first described and analyzed by Asai et al. [2]. They outlined several important Natural Language Processing (NLP) problems that can be studied using this data, such as discovering the activities that are most central to the happy moment, forming paraphrasings to describe those activities, creating an ontology of activities that cause happiness, and discovering whether the cause of happiness in a happy moment is related to the person's expectations. To demonstrate the level of diversity of the happy moments in the HappyDB corpus, Asai et al. [2] identified nine diverse topics (categories) that they saw occurring often in the corpus. These topics were labeled "people", "family" (subset of "people"), "pets", "work", "food", "exercise", "shopping", "school", and "entertainment". Those nine categories account for about 80% of the happy moments in the corpus, while the remaining 20% of the happy moments did not fit into any of those concepts and thus were gathered under the category "none".

The HappyDB data was used for the CL-Aff Shared Task which was organized as part of the 2nd Workshop on Affective Content Analysis at AAAI-19 [3]. The Shared Task was focused on using the HappyDB corpus for analyzing happiness and well-being in written language via the accomplishment of two sub-tasks: (1) Using a small labeled and large unlabeled training data to predict the two thematic labels "agency" and "sociality", where agency examines whether a particular emotion stems from the individual (self-caused), is inspired by other individuals (other-caused), or results solely from the situation (circumstance-caused), while sociality refers to whether or not people other than the author are involved in the emotion situation; (2) Developing interesting ways for automatic characterization of the happy moments in terms of affect, emotion, participants and content. The performance of the eleven teams participating in the Shared Task was compared based on their *Accuracy* and *F-score* at predicting the agency and sociality labels on an unseen test dataset that was collected following the same way as the original HappyDB data. Jaidka et al. [3] summarize the approaches followed by the participating teams as well as the results obtained by those teams. For example, Rajendran et al. [4] used deep neural networks and a variety of embedding methods to achieve accuracies of 87.97% and 93.13% at predicting agency and sociality, respectively. Syed et al. [5] obtained very similar results by utilizing a language model pre-trained on the WikiText-103 corpus using AWD-LSTM, a regularization of long short-term memory networks (LSTMs) [6]. Wu et al. [7] used syntactic, emotional and profile features as well as word embedding (Word embeddings are geometrical representations of words in dense vectors of a specific dimension. From these word vectors, things like word similarity can be computed, thus bringing differently worded but semantically similar documents together, which can be useful in document classification.) in logistic regression, Boosted Random Forests and Convolutional Neural Networks (CNN) to predict agency and sociality. Their best performing model, CNN with all four types of features, had F-scores of 0.80 and 0.90 for agency and sociality, respectively. They also extended their models to predict 15 categories labels (containing the nine categories defined in [2]) within the HappyDB corpus.

It is clear from the literature cited above that nearly all the work done on the HappyDB data so far was focused on predictive tasks. In this paper, we focus mainly on explaining the relationship between the social traits of the participants (i.e., age, gender, marriedhood, and parenthood) on one hand, and the "causes" and "expressions" of happiness on the other hand, to see to what extent these traits make a difference to how someone may feel happy. Although we also attempt to predict these traits from textual data, prediction is not our chief concern. We primarily seek to examine the semantic and lexical correlates of happiness as expressed by people of different ages, genders, and marital and parental statuses. For this aim, we utilize NLP tools (Part of Speech tagging, lemmatization, syntactic parsing and topic modeling), with the overall idea being that different groups feel and express happiness differently. The feeling is operationalized through discovering the causes of happiness,

and the expression through the analysis of their lexico-grammatical output. Then, we input the results obtained from applying these tools into a simple, but highly interpretable, classifier (logistic regression) to quantify the differences between the members of different social groups in terms of their causes and expressions of happiness.

The remainder of this paper is structured as follows. Section 2 introduces the HappyDB data and describes the methods used in our analyses. In Section 3, we report our explanatory results which provide answers to Questions (i–iii) listed above. The prediction results addressing Question (iv) are reported in Section 4. The paper is concluded by a discussion about the potential applications and extensions of our analyses.

## 2. Data and Methods

### 2.1. The HappyDB Data

In this study, we used the HappyDB dataset which was collected and made publicly available by Asai et al. [2]. This dataset is a corpus of more than 100,000 happy moments crowd-sourced via Amazon's Mechanical Turk. Each worker was asked to answer the question "*What made you happy in the past 24 h (or, alternatively, the past three months)?*" (Asai et al. [2]). The original HappyDB data, i.e., before data cleaning, was split evenly between these two reflection periods (24 h and three months). The following are two sample moments from the corpus:

- I popped into the local shop after a very long hard day at work to buy some tobacco and decided to buy a scratchcard with my change and won PS3000.
- When I called my internet provider today and found out I have another week before I have to pay my bill which makes things a whole lot easier right now.

The goal of this corpus is to advance the understanding of the causes of happiness through text-based reflection (Asai et al. [2]). Besides the happy moments listed by each individual, the data contain information about gender, age, marital status, parenthood status and country of residence for each individual.

Prior to any analysis, we performed some data cleaning which included omitting cases with entirely missing happy moments, removing repeated entries that had the same exact letter-by-letter text and demographic characteristics, and transforming some age values ($\leq 4227$ and 233 years) to missing. The cleaned data file contained exactly 99,930 cases. In Table 1 and Figure 1, we report some descriptive statistics from the final data.

**Table 1.** Sample distribution according to gender, marital status and parenthood status.

| Variable | Distribution | |
|---|---|---|
| Gender | Female (42%) | Male (58%) |
| Marital Status | Married (41%) | Unmarried (59%) |
| Parenthood Status | Parent (39%) | Non-parent (61%) |

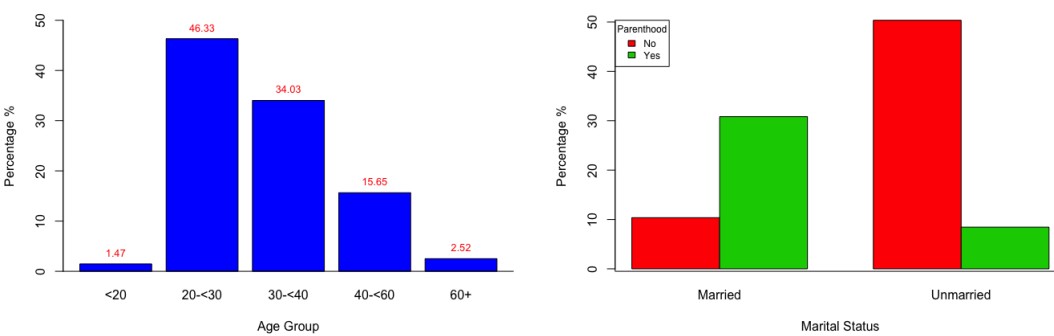

(**a**) Age distribution (Min = 17, Max = 98, Mean = 32.43, Median = 30).

(**b**) Marital status and parenthood.

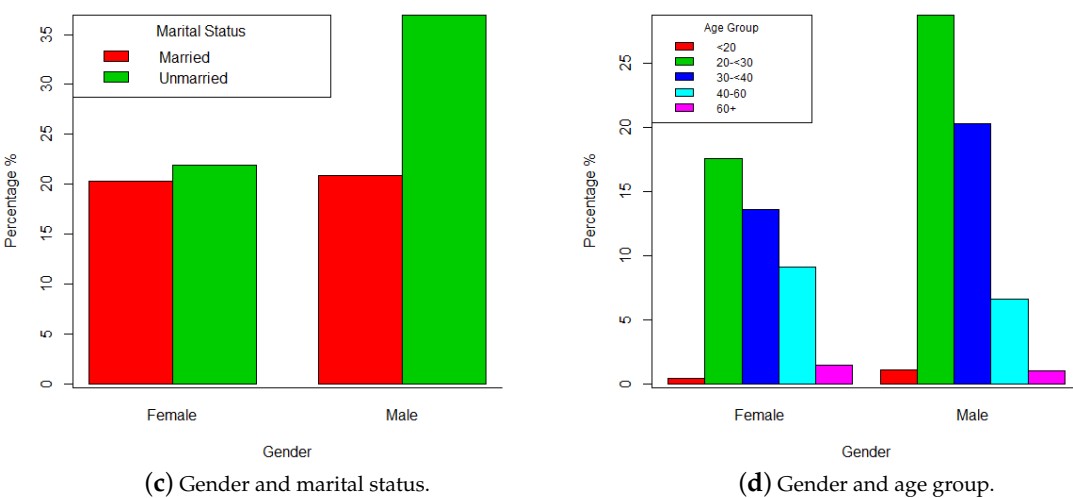

(**c**) Gender and marital status.

(**d**) Gender and age group.

**Figure 1.** (**a**–**d**) Sample distribution by different characteristics of the sample participants.

## 2.2. Topic Modeling

Topic modeling is an unsupervised machine learning technique by which a collection of documents is mapped to a number of topics. Each topic is a constellation of words, or in our case lemmas that share some semantic domain. In other words, topics are clusters of words that express a similar idea, and each document is modeled as a mixture of these topics. In the literature, there exist several algorithms for topic modeling such as Latent Semantic Analysis (LSA) [8,9], Probabilistic Latent Semantic Analysis (PLSA) [10], and Latent Dirichlet Allocation (LDA) [11]. In following, we briefly describe the steps involved in LDA, which is the most commonly used method for topic modeling. Suppose our corpus $\mathcal{C}$ consists of $D$ documents with document $d$ having $N_d$ words, $d = 1, \ldots, D$. LDA uses the following generative process to model the corpus $\mathcal{C}$ to $K$ topics [11]:

1. For document $d \in \{1, \ldots, D\}$, draw $\theta_d$ (topic proportion in the document) from the Dirichlet distribution with parameter $\alpha$.
2. For topic $k \in \{1, \ldots, K\}$, draw $\beta_k$ (topic distribution over the vocab) from the Dirichlet distribution with parameter $\eta$.
3. For a word $w_n (n \in \{1, \ldots, N_d\})$ in document $d$,

    (a) Draw a topic $z_n$ from the Multinomial distribution with parameter $\theta_d$.
    (b) Draw a word $w_n$ from the Multinomial distribution with parameter $\beta_{z_n}$.

In the above process, the topic distribution under each document is a Multinomial distribution, $Multi(\theta)$, with the Dirichlet conjugate prior $Dir(\alpha)$. The word distribution under each topic is $Multi(\beta)$ with conjugate prior $Dir(\eta)$. For the $n$-th word in a certain document, first draw a topic $z$ from the document-specific topic distribution $Mult(\theta)$, and then draw a word under this topic from the topic-specific word distribution $Mult(\beta)$. In addition, note that, in this process, the only observed variables are the words, whereas $z$, $\theta$ and $\beta$ are latent variables, and $\alpha$ and $\eta$ are hyperparameters. To make inferences about these latent variables and build the topics, the document-specific posterior distribution

$$P(\theta,\beta,\mathbf{z}|\mathbf{w},\alpha,\eta) = \frac{P(\theta,\beta,\mathbf{z},\mathbf{w}|\alpha,\eta)}{P(\mathbf{w}|\alpha,\eta)} = \frac{p(\theta|\alpha)\prod_{n=1}^{N}P(z_n|\theta)P(w_n|z_n,\beta)}{\int_\theta p(\theta|\alpha)\prod_{n=1}^{N}\sum_{z_n}P(z_n|\theta)P(w_n|z_n,\beta)\mathrm{d}\theta},$$

with $\mathbf{w} = (w_1,\ldots,w_N)$ and $\mathbf{z} = (z_1,\ldots,z_K)$, is approximated using the Variational Expectation-Maximization (VEM) algorithm [11] or Markov Chain Mote Carlo (MCMC) [12].

When topic modeling is applied to a corpus, we get a number of topics, predefined by the researcher, two of which may look like the following sample topics obtained from the application of LDA topic modeling to the HappyDB corpus:

- eat food lunch restaurant favorite pizza good dinner delicious order place local chinese great meal sushi today burger taste taco,
- trip vacation plan weekend friend summer book family visit week beach travel ticket flight girlfriend upcoming excited Florida holiday vegas

We can see that the words in each topic share a semantic association. For instance, it is quite obvious that the first sample topic is about dining, while the second one is about vacations. However, it should be noted that not all topics are going to be as clear as the ones listed above, and that sometimes we get topics that defy interpretation.

A fundamental step in topic modeling is the selection of the "optimal" number of topics to which the corpus should be mapped. In general, there are two ways to decide on the number of topics that one should use when running topic modeling. One way is to train topic modeling using different numbers of topics, say in the range of 2 to 100, and choose the number of topics that maximize or minimize certain metrics such as those discussed by Griffiths and Steyvers [12], Cao et al. [13] and Arun et al. [14]. The metric discussed in [12] results from adopting Bayesian model selection to determine the number of topics that best describes the structure of a given corpus. The main idea is to choose the model (specified by the number of topics) with the highest posterior probability given the data (the words in the corpus). This is equivalent to choosing the model given for which the likelihood of the data, $P(\mathbf{w}|K)$, is maximized since $P(\mathbf{w}|K)$ is the main component in the posterior probability of the model given the data. To avoid summing over all possible assignments of words to topics $\mathbf{z}$, they approximate $P(\mathbf{w}|K)$ by a harmonic mean of a set of values of $P(\mathbf{w}|\mathbf{z},K)$, where $\mathbf{z}$ is sampled from the posterior distribution $P(\mathbf{z}|\mathbf{w},K)$ using Gibbs sampling. Cao et al. [13] proposed a method for adaptive selection of the number of topics in LDA via clustering of topics based on topic density defined as the number of other topics falling within a certain radius from the topic under consideration. Their method is motivated by the argument that the optimal number of topics is correlated with the distances between topics, measured by the standard cosine distance. Given an initial number of topics $K_0$, the LDA model is estimated using the VEM algorithm. Then, the average cosine distance between the topics in the initial model is calculated and used as the radius to compute the density of each topic using which the cardinality of the model is obtained as the number of topics with density less than or equal to some threshold $n$. The model cardinality is then used to update the number of topics. This process is repeated until both the average cosine distance between topics and model cardinality stabilize. The selected number of topics is the one that minimizes the average cosine distance between the topics in the model. In [14], the measure proposed for selecting the number of topics is developed by viewing LDA as a matrix factorization method that factorizes a document-word frequency matrix

$M$ into two matrices $M_1$ (topic-word matrix) and $M_2$ (document-topic matrix) of dimension $K * N$ and $D * K$, respectively. The measure is then computed in terms of the symmetric Kullback–Leibler, divergence of the singular value distribution of $M_1$ and the distribution of the vector $LM_2$, where $L$ is a $1 * D$ vector containing the lengths of each document in the corpus. It was shown that the divergence values are higher for non-optimal number of topics, and, thus, the recommended number of topics is the one that minimizes the divergence measure. Teh et al. [15] and Zhao et al. [16] discuss other methods for selecting the number of topics in topic modeling.

Alternatively, the number of topics can be determined via manual inspection of a variety of topics sets trained using several different numbers of topics. When we manually examine the topics, we mainly look for the set of topics achieving *maximum coverage and minimum overlap*. In our analysis, we used both methods to decide on the number of topics to be used for the HappyDB corpus. We computed the three metrics reported in [12–14] using the function *FindTopicsNumber* in the R package *ldatuning* [17] by training several LDA models with the number of topics ranging from 2 to 100. The results displayed in Figure 2 suggest that the optimal number of topics with respect to these metrics is between 70 to 80 topics. Additionally, we ran topic modeling using $10, 20, \cdots, 100$ topics, and examined the resulting sets of topics manually. The result was almost unanimous preference for the word clusters produced based on the selection of 80 topics. Therefore, we decided to map the HappyDB corpus to 80 topics built using the LDA.

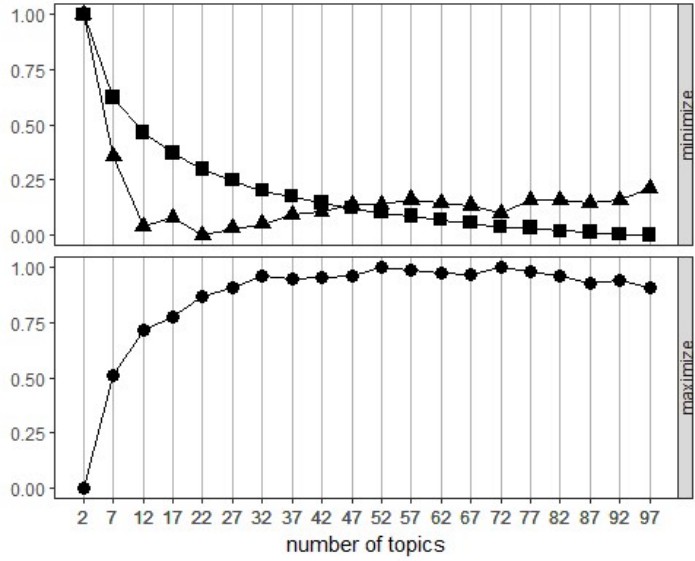

**Figure 2.** Selecting number of topics in topic modeling of the HappyDB corpus using three metrics: Griffiths2004 (●), CaoJuan2009 (▲), and Arun2010 (■).

We trained topic modeling in *MALLET (MAchine Learning for LanguagE Toolkit)* to build 80 topics summarizing the main ideas that are present in the corpus (see McCallum [18] for more details about MALLET). These topics are listed in Table A1 in the Appendix A. That table displays the topic number, topic label, topic weight and the top twenty keywords in each topic. The topics are sorted by their respective weights in the corpus. Topic labels classify each of the 80 topics generated by LDA into one of 18 unique categories based on the keywords of the topic. Those categories are a combination of the 10 categories defined by Asai et al. [2] and the 15 categories (concepts) used by Wu et al. [7], plus three new categories (see Section 1). The 18 categories are labeled "conversation", "education", "entertainment", "exercise", "family", "food", "gardening", "housekeeping", "money", "party", "pets", "romance","religion" "shopping", "vacation", "weather", "work" and "other". The category "other" contains the topics that do not clearly fit into any single category of the other 17 categories. The distribution of those categories over the 80 LDA topics is depicted in Figure 3, where the weight on the vertical access is calculated by summing up the weights of all topics belonging to the particular

category and dividing by the sum of weights of all 80 topics to make the categories weights add up to 1. From this figure, we see that the category "other" has weight 21% which is quite similar to the weight (20.3%) assigned to category "none" in [2]. Among the other 17 categories, "family" is the most dominant category in the HappyDB corpus, which is in agreement with the results reported in [2,7]. While in [2,7] "food" comes immediately after "family" followed by "work", "entertainment" and "shopping", LDA puts "work", "entertainment", "shopping" and "party" ahead of "food" as shown in Figure 3. Next, Asai et al. [2] list "exercise" followed by "education", Wu et al. [7] list "education" followed by "romance" and then "exercise", and LDA lists "education" followed by "exercise" and then "romance". Our ranking of the remaining categories is similar to the ranking of Wu et al. [7]. Finally, it should be noted that our categorization and the categorization in [2] are both based on the whole HappyDB dataset, whereas the categorization of Wu et al. [7] is based on only 10,560 cases.

This categorization of topics together with the results presented in the next section can be very useful in exploring whether the topics associated with males and females, married and unmarried people, parents and nonparents, and the people of different age groups, belong to the same or different categories (see Section 3.1 below).

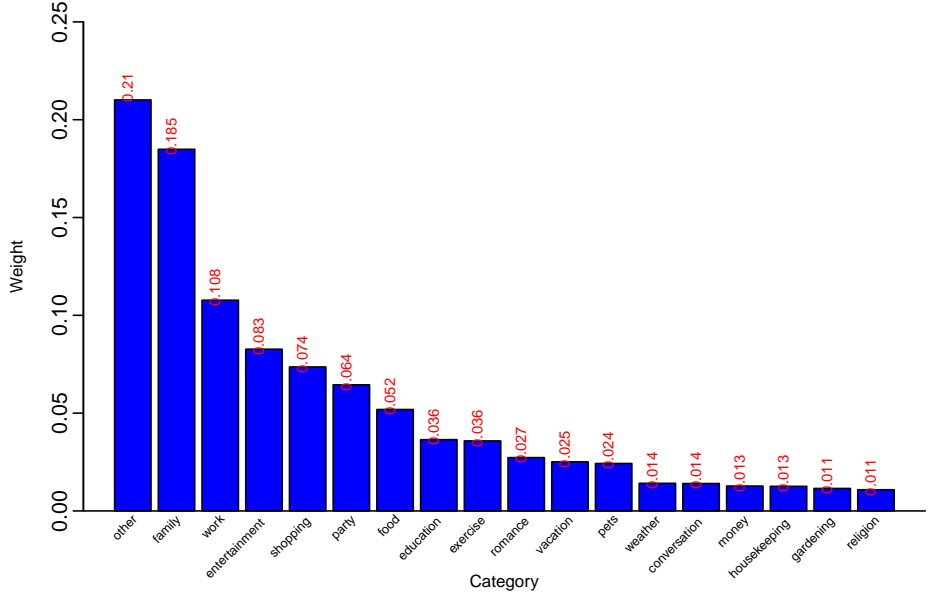

**Figure 3.** Distribution of categories over the LDA topics.

## 2.3. Lemmatization, Part of Speech Tagging, and Dependency Parsing

Words come in many different forms. For example, the word *play* can be seen in text as *plays*, *played*, and *playing*. It would be useful for an automatic system to know that all these words represent the same concept of *play*, the form which we usually find as the entry in dictionaries. Such a dictionary entry is called a *lemma*, and the process of finding the lemma of a specific word is called *lemmatization*.

Lemmatization by itself is usually enough to determine which lexical item one is dealing with, but we can also see that the word *play* can either be a noun or a verb, and the meanings of these can be quite different according to which *part of speech* (POS) the word represents. The process of automatically assigning parts of speech to words is called *part of speech tagging*, and is a standard process in computational linguistics. POS taggers for English usually have an accuracy of about 98%. For example, the sentence *The government will table the budget in the next meeting* can be assigned the tags *The/DET government/NN will/MOD table/VB the/DET budget/NN in/IN the/DET next/JJ meeting/NN*, where *DET* is a determiner, *NN* is a noun, *MOD* is a modal, *VB* is a verb and *JJ* is an adjective. The main role of POS tagging is thus grammatical disambiguation as the word *table*, which could be either a noun or a verb, is shown to be a verb in this specific context. It should be noted that POS tagging is required for correct lemmatization. The word *saw* can either be lemmatized as *see* or *saw* based on its POS tag.

*Dependency parsing* is a type of natural language syntactic analysis that treats grammatical relations as binary links between the words of a sentence. For example, in the sentence *I prefer the morning flight to Boston*, depicted in Figure 4, there is a binary relation between the verb *prefer* and the pronoun *I* of the type *nsubj* (nominal subject). There is also a binary relation of the type *dobj* (direct object) between the verb *prefer* and the noun *flight*. Other relations in the graph are *det* (determiner) and *nmod* (noun modifier).

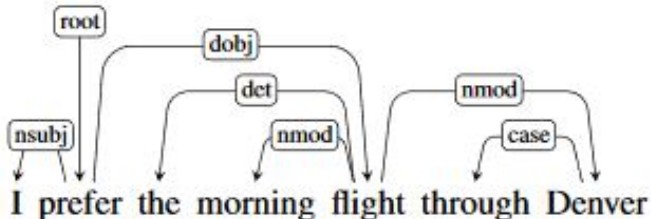

**Figure 4.** A sentence in dependency format (Source: [19] (p. 245)).

The point of using dependency parsing is that it re-arranges the sentences in a more semantically plausible way. If we use word bigrams as features, for example, we end up using: *I prefer, prefer the, the morning, morning flight, flight through, through Denver*. If we use dependency bigrams as features instead, we end up with: *I prefer, prefer flight, morning flight, flight through, through Denver*, thus catching the important relations between the verbs, subjects, and objects as well as all other grammatical relations. See [20] for a detailed discussion of dependency parsing.

For lemmatization, POS tagging, and dependency parsing, we used the *spaCy v2.0.11* package [21].

### 2.4. Predictive Models

In this section, we give a brief description of the algorithms used for predicting the different outcome variables (gender, marital status, parenthood status and age) from the text about happy moments. The results of these algorithms are presented in Section 4.

### 2.4.1. Logistic Regression

Logistic regression is one of the most famous classifiers in the statistics, data science, and machine learning worlds. For low-dimensional data, logistic regression is a standard approach for binary classification. This is especially true in scientific fields such as medicine, psychology, and social sciences where the focus is not only on prediction but also on explanation. There is also the multinomial version of logistic regression which can be used for modeling non-binary (multi-category) responses. Let us give a brief description of the binary logistic model. Let $Y$ be a binary response variable taking the values 0 and 1, and let $X_1, \ldots, X_p$ be a set of predictors (also called explanatory variables or features). Then, the logistic regression model takes the following form:

$$p := P(Y = 1 | X_1, \cdots, X_p) = \frac{\exp(\beta_0 + \beta_1 x_1 + \cdots + \beta_p x_p)}{1 + \exp(\beta_0 + \beta_1 x_1 + \cdots + \beta_p x_p)}, \tag{1}$$

where $\beta_0, \beta_1, \cdots, \beta_p$ are the regression coefficients. The maximum likelihood method is commonly used to obtain the estimated coefficients which are then substituted in Equation (1) to get the fitted model. Nonlinear least squares can also be used to fit the logistic model. The fitted model can then be utilized for future predictions. For instance, to predict the value of $Y$ for a future case having known $x_1, \cdots, x_p$ values, we basically plug-in these $x$-values and the estimated coefficients into the model in (1), and compute the estimated probability $\hat{p}$. Then, we predict $Y$ to be 1 for that case if $\hat{p}$ exceeds certain threshold $c$ ($c = 0.5$ is a common choice in practice), and predict $Y$ to be 0, otherwise.

The logistic model can be also written in the following form:

$$\log\left(\frac{p}{1-p}\right) = \beta_0 + \beta_1 x_1 + \cdots + \beta_p x_p, \tag{2}$$

where the left-hand-side is known as the *logit*, and the ratio $p/(1-p)$ is called the *odds*. In the context of logistic regression, the association between the outcome variable $Y$ and any of the predictors can be measured using the *odds ratio* (OR) which represents the odds of the event $(Y = 1)$ given a specific level for the predictor relative to the odds of the same event given a different level for the same predictor. For more details on odds ratios and logistic regression, see Agresti [22] (Sections 2.3 and 3.2) .

### 2.4.2. Gradient Boosting

Gradient boosting is a very powerful tree-based model that works via combing multiple decision trees which are classifiers that closely mirror the human decision-making process. Growing decision trees involves recursive binary splitting of the joint space of all predictors into a number of disjoint regions. A prediction for a given observation can then be made using the mean (regression trees with quantitative response) or the mode (classification trees with qualitative response) of the training observations in the region to which it belongs. Like logistic regression, decision trees are quite useful for interpretation. However, they typically do not have high prediction accuracy. Thus, the concepts of bagging, random forests, and gradient boosting were introduced as elegant ways to improve the prediction power of decision trees. The main idea behind these approaches rests in training multiple trees which are then combined to yield a single predictive model. In gradient boosting, these trees are grown sequentially, i.e., each tree is grown on a version of the original dataset modified using information from previously grown trees. The following is a description of the gradient boosting algorithm for the classification problem.

Suppose the response variable $Y$ has $K$ classes $\{\mathcal{G}_1, \ldots, \mathcal{G}_K\}$. Generalizing the logistic model in (1), the class conditional probabilities can be modeled as follows:

$$p_k(\mathbf{x}) := P(Y = \mathcal{G}_k | X_1, \cdots, X_p) = \frac{\exp\{f_k(\mathbf{x})\}}{\sum_{\ell=1}^{K} \exp\{f_\ell(\mathbf{x})\}}, \tag{3}$$

where $\mathbf{x} = (x_1, \ldots, x_p)$ and $\sum_{\ell=1}^{K} f_\ell(\mathbf{x}) = 0$ with the form of the functions $f_k$ being unknown. Alternatively, we can take $f_K(\mathbf{x}) = 0$ in Equation (3) to have the more familiar logistic model where the term "1+" appears in the denominator. The multinomial deviance loss function is commonly used in fitting the multi-class logistic model:

$$L(y, \mathbf{p}) = -\sum_{k=1}^{K} \mathbb{I}(y = \mathcal{G}_k) \log p_k(\mathbf{x}) = -\sum_{k=1}^{K} \mathbb{I}(y = \mathcal{G}_k) f_k(\mathbf{x}) + \log\left(\sum_{\ell=1}^{K} \exp\{f_\ell(\mathbf{x})\}\right),$$

where $\mathbb{I}(A)$ is the indicator function with value 1 if $A$ holds and value zero otherwise. Given this set-up, the gradient boosting algorithm involves the following steps:

1.  Set $f_{k0}(\mathbf{x}) = 0; k = 1, \ldots, K$.
2.  For $b = 1, \ldots, B$:

    (a)  Take $p_k(\mathbf{x})$ to be as in Equation (3).
    (b)  For $k = 1, \ldots, K$:

        i.   Compute $r_{ikb} = -\{\partial L(y, \mathbf{p})/\partial f_k(\mathbf{x}_i)\} = \mathbb{I}(y_i = \mathcal{G}_k) - p_k(\mathbf{x}_i); i = 1, \ldots, N$.
        ii.  Fit a regression tree with $m$ splits to the response $r_{ikb}, i = 1, \ldots, N$, producing terminal regions $R_{jkb}; j = 1, \ldots, m$.

iii. Compute the updating factors

$$\gamma_{jkb} = \{(K-1) \sum_{\mathbf{x}_i \in R_{jkb}} r_{ikb}\} / \{K \sum_{\mathbf{x}_i \in R_{jkb}} |r_{ikb}|(1-|r_{ikb}|)\}; j = 1, \ldots, m.$$

iv. Update $f_{kb}(\mathbf{x}) = f_{k(b-1)}(\mathbf{x}) + \sum_{j=1}^{d} \gamma_{jkb} \mathbb{I}(\mathbf{x} \in R_{jkb})$.

3. Output $\hat{f}_k(\mathbf{x}) = f_{kB}(\mathbf{x}); k = 1, \ldots, K.$

Usually using higher number of trees, $B$, results in better learning of the data, but it can also slow down the training process considerably. Therefore, a parameter search is needed to choose the appropriate $B$ and $m$. We refer the reader to Friedman [23] and Hastie et al. [24] (Chapter 10) for further details about gradient boosting and tree-based models in general.

### 2.4.3. FastText

FastText is a word representation and text classification library developed by Facebook's AI Research Lab. It uses neural networks and subwords as features in text representation and linear classifiers for text classification, and achieves state-of-the-art accuracy on common NLP datasets. The performance of this algorithm depends on few key parameters such as the learning rate and the number of epochs (the number of times each training example is seen by the algorithm) in training. Just as for gradient boosting, a parameter search is needed to tune the parameters of the fastText algorithm. For further details on fastText, see Joulin et al. [25].

### 3. Causes of Happiness, and Lexical and Grammatical Discrepancies

In this section, we provide answers to questions (i–iii) stated in Section 1. First, we use unsupervised topic modeling to explore the main causes of happiness for the participants of this study in general. As we mentioned in the previous section, the results of topic modeling are summarized in Table A1 in the Appendix A. These results shall be discussed in detail in this section. Second, we examine the similarities and dissimilarities, in terms of causes of happiness and the way of expressing and reporting these causes, among different social groups, such as males and females, parents and non-parents, married and unmarried people, and people from a variety of age groups. This examination is done by first transforming the happy moments into a set of lemmas, part of speech sequences, and dependency relations, in addition to the topics. Then, we use each set as predictors in multi-variable binary and multinomial logistic regressions to rank these predictors in terms of their influence on each outcome variable, namely, gender (female = 0, male = 1), parenthood status (non-parent = 0, parent = 1), marital status (unmarried = 0, married = 1), and age group (see Figure 1 for the list of age groups). Odds ratios resulting from the logistic regressions, hereafter called *scores*, are used to form this ranking. For instance, we consider a topic to be more common among females than males if it has a score less than or equal to 0.33, while it is considered more common among males if its score is greater than or equal to 3.00. These thresholds are commonly used in many areas of research such as psychological research [26]. In our analysis, this choice of the thresholds was also supported by observing that the 95% confidence intervals of the odds ratios do not contain 1.00, the point of indifference, whenever the observed score is outside the range $(0.33, 3.00)$.

### 3.1. Happiness and Gender

Here, we seek to answer the questions: which causes of happiness are more common among men versus women? What about words, lemmas, POS tags and dependency relations? We try to answer these questions using the steps outlined above. The results are presented below.

Topic models provide a semantic way to differentiate between the common causes of happiness for men and women. Out of the 80 topics, 16 show no significant difference between men and women (the confidence interval of the odds ratio contains 1). Of the remaining 64 topics, 21 topics seem to be

more feminine, and 43 seem to be more masculine. The top female topics and the top male topics are listed in Table 2. It is obvious from this table that there are different "causes" of happiness for men and women. While men are mostly interested in games and gadgets, women are more interested in family and friends. These differences continue to exist even when we compare the causes of happiness of males and females in terms of the more general categories discussed in Section 2.2. According to the topic labeling given in Table A1 in the Appendix A, the topics associated with males in Table 2 belong to the categories "entertainment" (2 topics), "shopping", "family" and "other", whereas the topics associated with females belong to the categories "shopping", "family", "pets", "education" and "party". Clearly, the topics associated with females cover more categories than those associated with males. Although both males and females talk about topics related to shopping and family, careful inspection of the topics themselves shows that the two groups talk about different kinds of shopping and different kinds of family-related matters.

**Table 2.** Top topics for males versus females.

| Males | | | Females | | |
|---|---|---|---|---|---|
| Topic | Score | Key Words | Topic | Score | Key Words |
| 73 | 22.24 | game play video friend fun beat online buy board finally nintendo level switch night enjoy hour zelda xbox pokemon win | 1 | 0.30 | happy time feel purchase open great find house box hand hold parent excited buy finally home thing day man door |
| 66 | 21.49 | game win team play watch baseball match basketball favorite soccer league son final score tournament playoff cricket ball hit beat | 46 | 0.33 | give son daughter hug big love smile kiss morning face year wake wife husband put baby home run pick kid |
| 17 | 19.56 | buy phone computer problem laptop fix work purchase figure mobile issue finally internet solve today system iphone cell save smartphone | 48 | 0.34 | cat bed great fall cuddle night sit dog fun lay colleague snuggle asleep sleep discussion couch wake lap morning love |
| 22 | 7.01 | life happiness people experience thing full part concept event live happy avoid show occasion tradition happen quality focus science seek | 27 | 0.39 | school college son graduate daughter year accept high student program teacher award class attend proud university receive degree summer scholarship |
| 34 | 6.75 | ago week month couple day year past hour girlfriend good finally back great wife start half happen completely marry end | 74 | 0.39 | gift day mother give birthday surprise husband mom wife card buy father present love flower happy special receive beautiful boyfriend |

The differences between men and women in this context are not only in the causes of happiness. There are also linguistic differences between female and male participants. When we use lemmas, rather than topics, as our predictor variables, we also notice that there is a disparate distribution for male and female lexical items. Table 3 lists the top female words and the top male words. It is realized that some of the words in Table 3 and the other tables of this section are not familiar English words. These words most likely represent names of places, games, shortcut of proper words, etc. For example, in Table 3, "tirupati" is a city in India, "stardew" is a video game, and "3mth" is a shortcut for "3 months".

Just as there are semantic (topic) differences between male and female texts, we also investigated the possibility of grammatical differences as represented by part of speech tags and dependency bigrams. As for linguistic differences in the syntactic structure, we can find differences using dependency syntax. For this purpose, we rely on the concept of a dependency triple, defined as two POS tags and the syntactic relation that holds between them. In the sentences, *The/DET woman/NN eats/VBZ pizza/NN*, there is a SUBJECT relation between the NN and the VBZ, and an OBJECT relation between the VBZ and the second NN. We can generalize these relations across all sentences and all POS tags to see whether there are gender differences among the distributions of these dependency triples. The results shown in Tables 4 and 5 demonstrate that indeed men and women differ in their use of syntactic structures when expressing their happy moments.

**Table 3.** Top words for males versus females.

| Males | | | | Females | | | |
|---|---|---|---|---|---|---|---|
| **Lemma** | **Score** | **Lemma** | **Score** | **Lemma** | **Score** | **Lemma** | **Score** |
| wife | 23.58 | girlfriend | 11.34 | husband | 0.03 | boyfriend | 0.06 |
| trading | 7.09 | tirupati | 6.56 | 3mth | 0.08 | hubby | 0.08 |
| gf | 5.93 | massive | 5.80 | oldage | 0.09 | makeup | 0.10 |
| elope | 4.96 | beggar | 4.93 | wonderla | 0.12 | blissful | 0.12 |
| restaurant | 4.88 | speculate | 4.85 | knit | 0.13 | crochet | 0.15 |
| seattle | 4.85 | football | 4.74 | children | 0.15 | necklace | 0.15 |
| gallon | 4.73 | bet | 4.70 | goodness | 0.15 | grandkid | 0.16 |
| biking | 4.63 | electronic | 4.53 | purse | 0.16 | sewing | 0.17 |
| furious | 4.52 | official | 4.49 | advertise | 0.17 | babysit | 0.18 |
| recovery | 4.46 | production | 4.39 | pedicure | 0.18 | stardew | 0.19 |

**Table 4.** Parts of speach sequences for males versus females.

| Males | | Females | |
|---|---|---|---|
| **Sequence** | **Score** | **Sequence** | **Score** |
| nnp nn prp nn cc | 5.530 | cd jj nns prp nn | 0.225 |
| vbd prp vbd jj nns | 3.750 | dt jj nn cc wdt | 0.230 |
| prp nn nn nn vbg | 3.640 | wrb prp vbd vbg jj | 0.250 |
| jj prp jj nn nn | 3.486 | prp vbd dt nns cc | 0.250 |
| vbg nn prp nns dt | 3.390 | prp vbd nnp nnp vb | 0.265 |
| prp nn vbg prp prp | 3.320 | vbg prp jj nn dt | 0.268 |
| jj prp vbd prp vb | 3.280 | nn nn rb rb prp | 0.270 |
| nnp dt nn rb | 3.250 | vbd prp vbd vbn dt | 0.280 |
| prp dt nn cc prp | 3.230 | rb vbd rp nn | 0.280 |
| vbn rp dt nn nn | 3.220 | prp nn nn vb dt | 0.285 |

**Table 5.** Dependency relations by gender.

| Males | | | | Females | | | |
|---|---|---|---|---|---|---|---|
| **HPOS** | **DPOS** | **DEP** | **Score** | **HPOS** | **DPOS** | **DEP** | **Score** |
| VBP | VBP | xcomp | 9.01 | VBG | VBD | csubj | 0.20 |
| NN | NNS | npadvmod | 5.34 | NN | NFP | punct | 0.22 |
| NN | PRP | dative | 5.04 | NNPS | NNP | amod | 0.24 |
| VBD | NNP | conj | 4.58 | IN | PRP$ | poss | 0.27 |
| VB | BNN | xcomp | 3.73 | PDT | RB | advmod | 0.28 |
| VBD | RB | conj | 3.67 | VBN | VBD | dep | 0.28 |
| VBD | UH | dobj | 3.43 | RP | NNS | npadvmod | 0.28 |
| LSB | -RRB- | punct | 3.21 | UH | NNS | pobj | 0.29 |
| VBN | VBD | parataxis | 3.20 | VBN | JJ | nsubjpass | 0.29 |
| JJ | VBZ | conj | 3.17 | VBD | ADD | punct | 0.29 |

*3.2. Happiness and Parenthood*

There are a lot of research studies the relationship between parenthood and happiness (or subjective well-being), e.g., Brenning et al. [27] and Vanassche et al. [28], but there does not seem to be enough work that differentiates parents from non-parents in terms of their *causes of happiness*. In this section, we try to bridge this gap by focusing on the topics, lexical items, and structures that set these two groups apart. Again, we use the approach outlined at the beginning of Section 3.

Table 6 compares the top topics among parents and non-parents. A careful look at these topics shows clearly that parents are first and foremost made happy by the well-being of their families and children. There is much focus on kids, kids' school success, and playing with kids. There is also mention of husbands and wives. Non-parents, on the other hand, are more interested in friends,

games, eating out, pets, and watching TV. It seems that these are two completely different worlds, or at least world-views.

**Table 6.** Top topics for parents versus nonparents.

| | Parents | | | Nonparents | |
|---|---|---|---|---|---|
| Topic | Score | Key Words | Topic | Score | Key Words |
| 57 | 11875 | son daughter year child young school learn kid watch play start pick picture excited make toy show proud grand dance | 49 | 0.04 | friend girl text send talk meet date girlfriend message picture good boyfriend facebook guy post crush woman cute love pretty |
| 46 | 110.70 | give son daughter hug big love smile kiss morning face year wake wife husband put baby home run pick kid | 73 | 0.05 | game play video friend fun beat online buy board finally nintendo level switch night enjoy hour zelda xbox pokemon win |
| 44 | 6.98 | baby happy sister birth child give bear movement time month wife pregnant day boy girl son wait moment person family | 3 | 0.10 | eat food lunch restaurant favorite pizza good dinner delicious order place local chinese great meal sushi taco today burger taste |
| 25 | 5.23 | fun play park lot kid enjoy time friend daughter day son fish pool water catch yesterday beach weekend swim family | 76 | 0.11 | gym good feel great today workout hair work morning exercise personal yoga start cut record lift session class run haircut |
| 27 | 3.74 | school college son graduate daughter year accept high student program teacher award class attend proud university receive degree summer scholarship | 13 | 0.11 | friend meet good time long hang year talk mine catch close school chat college lunch childhood visit great fun bar |

In Table 7, we report the words that parents use the most and the words that are more common among non-parents. The top parents' words are obviously family-related and show that we could actually be dealing with both parents and grandparents, e.g., *daughter, son, grandson, granddaughter, grandchild, grandkid, kid*. On the contrary, top words for non-parents are a bit harder to group under one category, but obviously show interest in games, examinations and movies.

**Table 7.** Top words for parents versus nonparents.

| **Parents** | | | | **Nonparents** | | | |
|---|---|---|---|---|---|---|---|
| **Lemma** | **Score** | **Lemma** | **Score** | **Lemma** | **Score** | **Lemma** | **Score** |
| daughter | 66.10 | son | 56.23 | laws | 0.08 | marks | 0.11 |
| grandson | 25.20 | granddaughter | 22.76 | toys | 0.12 | orphan | 0.13 |
| grandchild | 22.10 | kid | 13.70 | anime | 0.17 | heroes | 0.17 |
| oldage | 10.15 | toddler | 9.13 | roommate | 0.19 | mumbai | 0.20 |
| chemo | 8.51 | child | 7.00 | brother | 0.20 | videogame | 0.20 |
| bhk | 6.40 | grandkid | 6.33 | lemonade | 0.20 | sir | 0.21 |
| sitt | 6.10 | maternity | 6.00 | midterm | 0.21 | neighborhood | 0.21 |
| children | 5.90 | stepson | 5.35 | bully | 0.22 | exclusive | 0.22 |
| helped | 5.25 | kiddo | 5.10 | russian | 0.22 | courage | 0.22 |
| feeder | 4.28 | babysitter | 4.22 | girlfriend | 0.22 | games | 0.23 |

### 3.3. Happiness and Marriage

Although people talk a lot about the joys of being single, research shows that married people tend to be happier than unmarried people, at least for a few years after marriage. When couples divorce, they tend to be less happy, and when people re-marry, their happiness goes up (e.g., Veenhoven [29]).

In this section, we are interested, not in the impact of marriage on happiness, but in the relationship between marriage and causes of happiness. We particularly seek to answer the question "Do married people have different causes of happiness than unmarried people?"

The top topics for married and unmarried people are listed in Table 8. This table shows clear differences in the causes of happiness for the two social groups. While unmarried people focus mostly on dating, friendship, food, and workout, married people are mainly focused on children and family bonding and events. This result is in agreement with the fact that most of the participants who are married are also parents, while most of the unmarried participants are non-parents as displayed in Figure 1b. This difference can also be shown in lexical items as presented in Table 9.

**Table 8.** Top topics for married versus unmarried individuals.

| | Married | | | Unmarried | |
|---|---|---|---|---|---|
| Topic | Score | Key Words | Topic | Score | Key Words |
| 57 | 126.15 | son daughter year child young school learn kid watch play start pick picture excited make toy show proud grand dance | 49 | 0.02 | friend girl text send talk meet date girlfriend message picture good boyfriend facebook guy post crush woman cute love pretty |
| 46 | 31.85 | give son daughter hug big love smile kiss morning face year wake wife husband put baby home run pick kid | 73 | 0.11 | game play video friend fun beat online buy board finally nintendo level switch night enjoy hour zelda xbox pokemon win |
| 44 | 5.31 | baby happy sister birth child give bear movement time month wife pregnant day boy girl son wait moment person family | 13 | 0.13 | friend meet good time long hang year talk mine catch close school chat college lunch childhood visit great fun bar |
| 37 | 3.57 | make happy mother sister father lunch law time son home wife roti part daughter morning serve brother leave learn cook | 3 | 0.15 | eat food lunch restaurant favorite pizza good dinner delicious order place local chinese great meal sushi taco today burger taste |
| 32 | 3.20 | temple family church easter enjoy yesterday god festival people sunday service member egg attend kid volunteer pray hunt trump morning | 76 | 0.16 | gym good feel great today workout hair work morning exercise personal yoga start cut record lift session class run haircut |

**Table 9.** Top words for married versus unmarried individuals.

| | Married | | | | Unmarried | | |
|---|---|---|---|---|---|---|---|
| Lemma | Score | Lemma | Score | Lemma | Score | Lemma | Score |
| husband | 35.53 | wife | 20.86 | monopoly | 0.07 | boyfriend | 0.07 |
| attended | 10.48 | hubby | 10.26 | fiancee | 0.08 | divorce | 0.11 |
| oldage | 9.89 | spouse | 9.64 | dawn | 0.14 | girlfriend | 0.15 |
| tempel | 7.31 | persistance | 6.95 | midterm | 0.16 | citation | 0.16 |
| bhk | 6.90 | needy | 6.48 | du | 0.18 | monitor | 0.18 |
| wonderla | 6.15 | seedling | 6.02 | roommate | 0.18 | mistakenly | 0.19 |
| daughter | 5.68 | pit | 5.63 | 3mth | 0.19 | toys | 0.19 |
| nowadays | 5.60 | helpd | 5.55 | life | 0.19 | agency | 0.19 |
| son | 5.39 | toddler | 5.28 | sir | 0.20 | laws | 0.21 |
| romantic | 5.23 | wend | 5.23 | enthusiasm | 0.21 | custody | 0.22 |

*3.4. Happiness and Age*

One of the major questions this study is trying to answer is the relationship between age and happiness, not in terms of how age affects happiness per say, but in terms of how age difference shapes our understanding of happiness. We examine how causes of happiness, as represented in topics, vary by age. We study the association between age and happiness by testing if we can use individuals' statements about happiness to classify these individuals into different age groups. Therefore, the study participants were divided into five age groups: Group0 (<20 years old); Group1 (20 $\leq$ 30 years old); Group2 (30 $\leq$ 40 years old); Group3 (40 $\leq$ 60 years old); Group4 (60+ years old). When defining these groups, we were hoping to gather individuals experiencing similar stages in their life in the same group and at the same time maintain a reasonable balance between the sizes of the groups. The distribution of participants among these five age groups is depicted in Figure 1a.

For logistic regression, we chose Group1 as our base category since this is the dominating age group in the dataset (46.33%). Thus, our results will compare all of the four other age groups to Group1.

Table 10 displays the top topics for each of the four age groups relative to Group1. From this table, it is clear that the causes of happiness for Group0 (<20 years old) are quite different from those for all other groups. Clearly, the causes of happiness for Group0 center around progress in school or work. Furthermore, we notice that the other three age groups (30+ years old) have similar causes of happiness with Topics 42, 57 and 65 being three top topics shared by these groups. Group2 and Group3 share another top topic, Topic 46, which is mainly about kids or family in general, while the fourth major topic among Group4 is Topic 58 which is mainly about enjoying/describing weather conditions.

**Table 10.** Top topics for different age groups relative to Group1 ($20 \leq 30$ years old).

| Topic | Score | Key Words | Topic | Score | Key Words |
|---|---|---|---|---|---|
| **Age < 20** | | | **$30 \leq$ Age $< 40$** | | |
| 47 | 42.47 | class pass exam test grade finish good final semester college study score hard high son school student receive daughter paper | 57 | 50.40 | son daughter year child young school learn kid watch play start pick picture excited make toy show proud grand dance |
| 70 | 12.84 | happy make event husband recently feel week extremely girlfriend love big small incredibly excited lastly fulfil significant super spouse plan | 46 | 7.27 | give son daughter hug big love smile kiss morning face year wake wife husband put baby home run pick kid |
| 21 | 9.90 | happy make month past event hour thing happen weekend small girlfriend occur big recent learn mth fiance involve twenty hrs | 65 | 4.27 | plant garden yard flower start tree air grow lawn rain water nice bloom smell spring mow seed finally grass bath |
| 19 | 6.30 | make happy feel today yesterday proud thing work progress turn love successful compliment real significant april pretty decision point sincerely | 42 | 3.18 | doctor find back surgery good hospital health finally pain cancer dog week appointment sick recover blood news today test vet |
| **$40 \leq$ Age $< 60$** | | | **Age $\geq 60$** | | |
| 57 | 161.92 | son daughter year child young school learn kid watch play start pick picture excited make toy show proud grand dance | 57 | 386.56 | son daughter year child young school learn kid watch play start pick picture excited make toy show proud grand dance |
| 65 | 25.45 | plant garden yard flower start tree air grow lawn rain water nice bloom smell spring mow seed finally grass bath | 65 | 81.00 | plant garden yard flower start tree air grow lawn rain water nice bloom smell spring mow seed finally grass bath |
| 42 | 8.93 | doctor find back surgery good hospital health finally pain cancer dog week appointment sick recover blood news today test vet | 42 | 31.45 | doctor find back surgery good hospital health finally pain cancer dog week appointment sick recover blood news today test vet |
| 46 | 6.90 | give son daughter hug big love smile kiss morning face year wake wife husband put baby home run pick kid | 58 | 14.22 | walk weather day nice rain warm today sun beautiful sunny enjoy spring bird cool long hot sit dog morning cold |

## 4. Prediction

We have so far tackled Questions (i–iii), stated in Section 1, about the causes of happiness in general and how different social groups diverge in terms of their causes of happiness and the way of expressing happiness. Another equivalently important question is whether one, given a piece of text, can predict whether that piece was written by a man or a woman, a married or an unmarried person, a parent or a non-parent, a young or an old person. This question falls under the umbrella of document classification research, and we treat it as such.

For the prediction of the different outcome variables considered in this paper, we employed three classification algorithms, namely, logistic regression as a simple and easily interpretable predictive algorithm, gradient boosting as a tree-based classification algorithm known to have excellent predictive power, and fastText as a deep-learning algorithm. These algorithms are briefly described in Section 2.4. Each of these algorithms was trained using some of the following sets of predictors:

1. *Dependency Bigrams*. In this experiment, we re-arrange the order of the document in terms of dependencies. For example, the sentence *I like hot pizza and Elizabeth likes cold pizza* becomes

    {(like, I), (like, Elizabeth), (like, pizza), (pizza, hot), (pizza, cold)}. We use dependencies as binary features with 1 if they occur in a document, and 0, otherwise.

2.  *Lemmas*. In this experiment, all words are lemmatized, and the features are unigrams to trigrams.
3.  *Word Forms*. In this experiment, the original words of the document are used as is, without any lemmatization, with the features being unigrams to trigrams.
4.  *Topics*. In this experiment, each document is converted to topic features where each document is 80 features, with each being the probability of the document belonging to each of the 80 topics built by Mallet.
5.  *Character Ngrams*. In this experiment, we use only characters (including spaces) as features with the features being unigrams to five grams.

The prediction results of logistic regression models for predicting gender, parenthood status, and marital status are reported in Table 11. In this table, we present the values of three commonly used accuracy measures; *Precision, Recall* and *F-score*. These measures are computed from the test data which represents 20% of the whole dataset, while the models were trained using the remaining 80% of the data. It is readily seen from Table 11 that the topics, although very useful for explanation purposes as seen in the previous section, are not the best predictors for any of the three outcome variables: gender, parenthood status, and marital status. We can also see that the other predictor sets are similar in their predictive power with dependency bigrams and character ngrams providing more or less equal results. Character ngrams do not use words, but letters, including spaces. This indicates that advanced linguistics techniques like POS tagging, lemmatization, and dependency parsing, while very useful in explanation, are not as useful in prediction, and we can obtain decent results without recourse to any advanced text processing.

While logistic regression may give a good baseline, other more recent algorithms, such as fastText and gradient boosting, have proven very successful in text classification. These two algorithms are best known to give high accuracy in text classification. In Table 12, we summarize the results from using these algorithms for predicting gender, marital status, parenthood status, and age.

We ran non-comprehensive grid searches to tune the parameters of the fastText and the gradient boosting algorithms. The data was first split into three sets: training 75%, validation 10%, and test 15%. The validation data was used for parameter tuning. For fastText, we tested ngrams from 1 to 4, epochs in the range of 10, 50, 100, 500 and 1000, and learning rates from 0.1 to 1 with 0.1 increments. For gradient boosting, we tested 3 to 10 character ngrams, 1 to 3 word ngrams, learning rates from 0.1 to 1 with 0.1 increments, depths from 3 to 30, number of trees in the range of 10, 100, 300 and 1000, and whether term frequency-inverse document frequency (tf-idf) should be used. The best performing features for fastText turned out to be the use of unigrams, rather than higher order ngrams, with 10 training epochs. The learning rate varied among categories with the best being 0.9 for parenthood status, 0.8 for gender, and 0.1 for the rest. The use of external word embeddings did not result in any significant improvements in the fastText predictions. On the other hand, the best performing settings and features for gradient boosting were a learning rate of 0.1, a maximum tree depth of 3, the number of trees being 1000, and with the utilization of tf-idf and a combination of word unigrams and bigrams.

From Table 12, we notice the following: parenthood status is the least difficult to predict, which may be interpreted as that being a parent is a good indicator of happiness or simply that parents often talk about being parents whether this lead to happiness or not. Gender is next on the list, and as we have seen in topic modeling, there are distinctive male and female causes of happiness. A similar conclusion holds true for marital status. Again, there seem to be specific indicators that set married and unmarried people apart. While the data does not tell us which ones are happier, it tells us that there is a difference. Predicting age was the hardest of all categories. We used multi-variable linear regression with age as a continuous response, and multinomial logistic regression, fastText and gradient boosting with age group as a categorical response. The results of linear regression and logistic regression for predicting age were not satisfactory at all, while the age prediction results from fastText

and gradient boosting were below average as shown in Table 12. This result may be understood as that either people do not talk enough about their age when expressing their happy moments, hence the difficulty of finding lexical indicators, or that happiness cannot be easily associated with age, a claim we do not have the capacity to hold or deny.

**Table 11.** Predicting gender, parenthood status, and marital status using various sets of predictors in logistic regression.

| Predictors | | Gender | | | | Parenthood | | | | Married | | |
|---|---|---|---|---|---|---|---|---|---|---|---|---|
| | | Prec. | Rec. | F | | Prec. | Rec. | F | | Prec. | Rec. | F |
| **Topics** | Male | 0.61 | 0.87 | 0.72 | Yes | 0.65 | 0.20 | 0.30 | Yes | 0.60 | 0.21 | 0.31 |
| | Female | 0.55 | 0.23 | 0.32 | No | 0.64 | 0.93 | 0.76 | No | 0.61 | 0.90 | 0.73 |
| | Total | 0.59 | 0.60 | 0.55 | Total | 0.64 | 0.64 | 0.58 | Total | 0.61 | 0.61 | 0.55 |
| **Dependecy Bigrams** | Male | 0.69 | 0.81 | 0.75 | Yes | 0.68 | 0.52 | 0.59 | Yes | 0.65 | 0.53 | 0.58 |
| | Female | 0.66 | 0.50 | 0.57 | No | 0.72 | 0.84 | 0.78 | No | 0.70 | 0.80 | 0.75 |
| | Total | 0.68 | 0.68 | 0.67 | Total | 0.70 | 0.71 | 0.70 | Total | 0.68 | 0.69 | 0.68 |
| **Lemmas (1–3 grams)** | Male | 0.70 | 0.80 | 0.74 | Yes | 0.71 | 0.88 | 0.79 | Yes | 0.71 | 0.88 | 0.79 |
| | Female | 0.65 | 0.52 | 0.58 | No | 0.72 | 0.45 | 0.55 | No | 0.72 | 0.45 | 0.55 |
| | Total | 0.68 | 0.68 | 0.67 | Total | 0.71 | 0.71 | 0.69 | Total | 0.71 | 0.71 | 0.69 |
| **Word Forms (1–3 grams)** | Male | 0.70 | 0.80 | 0.74 | Yes | 0.69 | 0.47 | 0.56 | Yes | 0.70 | 0.50 | 0.58 |
| | Female | 0.65 | 0.52 | 0.58 | No | 0.71 | 0.86 | 0.78 | No | 0.70 | 0.85 | 0.77 |
| | Total | 0.68 | 0.68 | 0.67 | Total | 0.70 | 0.71 | 0.69 | Total | 0.70 | 0.70 | 0.69 |
| **Char Ngrams (1–5 grams)** | Male | 0.69 | 0.82 | 0.75 | Yes | 0.66 | 0.52 | 0.58 | Yes | 0.63 | 0.55 | 0.59 |
| | Female | 0.67 | 0.50 | 0.57 | No | 0.72 | 0.82 | 0.77 | No | 0.70 | 0.77 | 0.74 |
| | Total | 0.68 | 0.69 | 0.68 | Total | 0.69 | 0.70 | 0.69 | Total | 0.67 | 0.68 | 0.67 |

**Table 12.** Predicting gender, parenthood status, marital status, and age group using fastText and gradient boosting.

| Resopnse | FastText | | | Gradient Boosting | | |
|---|---|---|---|---|---|---|
| | Prec. | Rec. | F | Prec. | Rec. | F |
| **Gender** | 0.669 | 0.669 | 0.669 | 0.680 | 0.670 | 0.675 |
| **Parenthood Status** | 0.703 | 0.703 | 0.703 | 0.736 | 0.721 | 0.728 |
| **Marrital Status** | 0.659 | 0.659 | 0.659 | 0.667 | 0.667 | 0.667 |
| **Age Group** | 0.491 | 0.491 | 0.491 | 0.476 | 0.498 | 0.487 |

## 5. Discussion

In this article, we presented empirical results investigating the general causes of happiness for different social groups and how the expression of happiness could be different for these groups. We demonstrated that each of these social groups has its own lexical items. It is tempting to unify this lexical item specialization in its own framework, which gives rise to our idea of *SocioWordNet*, a lexical database of English that assigns for each lexical item how it is used by different genders, married and unmarried people, parents and non-parents, and people of different ages. This kind of *Enriched Lexical Profiling* could be useful in many linguistic as well as industrial applications: socio-linguistic research, automatic conversational agents, sentiment analysis, and basically any linguistic task where inference is of essence.

In Table 13, each word is scored across three variables, with the scores being the odds ratio of the word in this specific context. To put things in perspective, the words *beautiful*, *pretty*, *gorgeous*, *handsome*, and *lovely* are more or less synonymous, but they are also used by different kinds of people with varying degrees. The idea behind enriched lexical profiling is that we can use these features to build linguistic resources with this specific, and more, information. This could be used in such applications as dialogue systems, user profiling, and socio-linguistic and cultural research. We are

currently working on building such *SocioWordNet*. We have so far collected a 200 million word corpus with explicit and implicit sociological indicators.

**Table 13.** Examples of enriched lexical profiling (the numbers are odds ratio scores).

| Lemma | Parent | Married | Male | Lemma | Parent | Married | Male |
|---|---|---|---|---|---|---|---|
| school | 0.99 | 1.10 | 1.03 | beautiful | 1.28 | 1.28 | 0.61 |
| girlfriend | 0.22 | 0.01 | 11.34 | pretty | 0.72 | 0.35 | 1.18 |
| boyfriend | 0.51 | 0.00 | 0.06 | gorgeous | 0.88 | 1.68 | 1.67 |
| xbox | 0.59 | 0.33 | 2.03 | handsome | 1.39 | 1.54 | 1.90 |
| football | 1.50 | 0.68 | 4.70 | lovely | 0.81 | 2.31 | 0.84 |
| friend | 0.64 | 0.18 | 1.04 | | | | |
| happy | 1.45 | 3.42 | 0.74 | | | | |
| happiness | 0.93 | 1.83 | 0.96 | | | | |
| precious | 0.69 | 1.32 | 0.19 | | | | |
| blissful | 0.88 | 0.73 | 0.12 | | | | |

We have shown in this paper that the *causes of happiness* can be extracted from textual data through the utilization of semantic tools (i.e., topic modeling) and linguistic tools (i.e., distinctive vocabulary lists). We have also shown that computational linguistics can provide us with a finer analysis that highlights the differences between genders, parents and non-parents, married and unmarried people, and the different age groups. Although we have also shown that gender, parenthood status, marital status, and age can in principle be predicted from textual data, with varying degrees of success, our goal was mainly exploratory rather than predictive.

We see the value of this work in the doors it opens for future research topics such as building a sociologically-enriched lexicon of English, studying the causes of happiness through Bayesian Belief Networks, and merging linguistic and non-linguistic data for the study of human subjective well-being. Perhaps, the most salient implication of this study is the possibility that happiness may be couched in quantitative terms. While this paper describes what people think is happiness, can reversing the process, through custom-tailoring components of happiness for specific groups of people, actually produce happiness? We personally believe that happiness is much more than a number of components mixed together, and that the question is actually too big for us to tackle. Research into creating the perfect recipes for happiness definitely requires the collaboration of psychologists, medical professionals, linguists, statisticians, artificial intelligence researchers and many others. We certainly wish such recipes could be found out.

Finally, it is worthwhile mentioning that this work is not without limitations. For starters, the data we have used is limited in size. Additionally, the data itself had some quality issues since some of it is written in a Twitter-like fashion that requires improving our tools or working on correcting the orthography to work with the current tools. We have also not examined the interactions among the several outcome variables in our dataset. These points shall be addressed in our future research.

**Author Contributions:** Conceptualization, E.M. and S.M.; Formal analysis, E.M. and S.M.; Methodology, E.M. and S.M.; Writing—original draft, E.M. and S.M.; Writing—review & editing, E.M. and S.M.

**Funding:** This research received no external funding.

**Acknowledgments:** The authors are indebted to the Editor and four anonymous referees for their insightful suggestions that led to improving this paper substantially.

**Conflicts of Interest:** The authors declare no conflict of interest.

## Appendix A. Results of Topic Modeling

In this appendix, we report the results of topic modeling. More specifically, in Table A1 below, we list the top twenty keywords in each of the eighty topics to which the corpus of happy moments has been mapped. In that table, the topics are sorted by their respective weights in the corpus.

**Table A1.** Eighty topics sorted by their weights in the corpus.

| Topic | Label | Weight | Key Words |
|-------|-------|--------|-----------|
| 13 | Other | 0.06486 | friend meet good time long hang year talk mine catch close school chat college lunch childhood visit great fun bar |
| 21 | Other | 0.06374 | happy make month past event hour thing happen weekend small girlfriend occur big recent learn mth fiance involve twenty hrs |
| 30 | Other | 0.06019 | happy feel make good today yesterday sick pretty great morning lot mood care relaxed accomplished content relieved pleased bit comfortable |
| 62 | Family | 0.05937 | visit brother sister family live parent home house town year weekend mom cousin back city friend aunt state week month |
| 0 | Work | 0.05917 | work day today time long week home weekend relax break spend hour yesterday nice friday schedule afternoon boyfriend busy lot |
| 12 | Vacation | 0.05905 | trip vacation plan weekend friend summer book family visit week beach travel ticket flight girlfriend upcoming excited florida holiday vegas |
| 5 | Romance | 0.05899 | night wife girlfriend dinner nice date time boyfriend husband great spend evening sex kid romantic surprise anniversary partner enjoy saturday |
| 70 | Family | 0.05882 | happy make event husband recently feel week extremely girlfriend love big small incredibly excited lastly fulfil significant super spouse plan |
| 38 | Family | 0.05873 | time long spend family friend member visit enjoy great yesterday find drive house fun period distance girlfriend hard start laugh |
| 36 | Work | 0.05865 | work today home early leave day husband office hour find coworker boss lunch tomorrow earlier yesterday key late week wife |
| 59 | Work | 0.05856 | work receive job promotion boss raise give promote company good hard high compliment bonus salary manager performance meeting review employee |
| 58 | Weather | 0.05854 | walk weather day nice rain warm today sun beautiful sunny enjoy spring bird cool long hot sit dog morning cold |
| 72 | Conversation | 0.05807 | talk friend call phone speak hear good time hour nice sister catch long year mom conversation live brother chat uncle |
| 60 | Family | 0.05769 | dinner family cook meal eat nice husband favorite night home wife food restaurant prepare enjoy delicious mom dish tonight mother |
| 3 | Food | 0.05750 | eat food lunch restaurant favorite pizza good dinner delicious order place local chinese great meal sushi taco today burger taste |
| 74 | Party | 0.05742 | gift day mother give birthday surprise husband mom wife card buy father present love flower happy special receive beautiful boyfriend |
| 57 | Family | 0.05739 | son daughter year child young school learn kid watch play start pick picture excited make toy show proud grand dance |
| 26 | Party | 0.05703 | birthday friend party celebrate family surprise daughter son celebration invite brother yesterday house fun cake people throw lot gift enjoy |
| 31 | Work | 0.05691 | work project finish complete task week successfully month difficult finally assignment big today client large art accomplish long hard boss |

| Topic | Label | Weight | Key Words |
|---|---|---|---|
| 19 | Other | 0.05673 | make happy feel today yesterday proud thing work progress turn love successful compliment real significant april pretty decision point sincerely |
| 35 | Shopping | 0.05660 | find store buy grocery sale sell item money shopping deal stock price shop dollar good lot market save ebay purchase |
| 25 | Party | 0.05637 | fun play park lot kid enjoy time friend daughter day son fish pool water catch yesterday beach weekend swim family |
| 64 | Family | 0.05626 | happy make moment feel mom yesterday month today back day home dad everyday excited special enjoy ready extremely wait lot |
| 33 | Work | 0.05620 | job interview offer company work call receive position apply pay hire promotion good opportunity month accept select week manager career |
| 67 | Family | 0.05619 | time spend family day make weekend happy entire year kid quality home lot sunday plan house life child grandma mother |
| 34 | Family | 0.05612 | ago week month couple day year past hour girlfriend good finally back great wife start half happen completely marry end |
| 6 | Other | 0.05611 | sleep night morning wake early hour good bed today breakfast rest feel late nap full refresh kid saturday put alarm |
| 20 | Food | 0.05574 | make dinner eat chicken delicious cook good breakfast recipe lunch fry turn cheese wife great meal grill night egg healthy |
| 11 | Pets | 0.05547 | dog cat puppy walk play pet adopt park home run cute neighbor kitten animal shelter bring greet love feed rescue |
| 46 | Family | 0.05509 | give son daughter hug big love smile kiss morning face year wake wife husband put baby home run pick kid |
| 9 | Party | 0.05467 | friend wedding marriage family day function attend sister cousin happy celebrate marry party anniversary invite enjoy member brother year lot |
| 41 | Entertain-ment | 0.05417 | watch show tv favorite episode season video netflix youtube night television funny series enjoy find binge start comedy finale channel |
| 27 | Education | 0.05414 | school college son graduate daughter year accept high student program teacher award class attend proud university receive degree summer scholarship |
| 45 | Work | 0.05413 | job finally start time work find year week month business part end leave decide husband back full quit search call |
| 49 | Romance | 0.05401 | friend girl text send talk meet date girlfriend message picture good boyfriend facebook guy post crush woman cute love pretty |
| 73 | Entertain-ment | 0.05322 | game play video friend fun beat online buy board finally nintendo level switch night enjoy hour zelda xbox pokemon win |
| 79 | Money | 0.05276 | pay money bill tax credit card account bank month save loan finally debt check receive refund large payment amount extra |
| 47 | Education | 0.05260 | class pass exam test grade finish good final semester college study score hard high son school student receive daughter paper |
| 4 | Shopping | 0.05253 | buy find pair shopping shoe dress purchase clothe shop wear mall fit shirt sale good nice great store price deal |
| 50 | Work | 0.05242 | work mturk make hit money bonus receive today extra pay earn amount yesterday turk goal mechanical amazon survey complete dollar |

**Table A1.** *Cont.*

| Topic | Label | Weight | Key Words |
|---|---|---|---|
| 28 | Other | 0.05241 | move house apartment find live home finally place city year wife decide rent back boyfriend close buy state neighbor roommate |
| 44 | Family | 0.05228 | baby happy sister birth child give bear movement time month wife pregnant day boy girl son wait moment person family |
| 42 | Other | 0.05228 | doctor find back surgery good hospital health finally pain cancer dog week appointment sick recover blood news today test vet |
| 55 | House-keeping | 0.05216 | clean house room finally paint finish kitchen put husband laundry home yesterday living bathroom entire organize wash buy apartment chore |
| 54 | Shopping | 0.05212 | car buy drive fix purchase finally find repair save break truck gas tire month vehicle brand wash pay pick change |
| 16 | Family | 0.05178 | year finally month week time ago love start back day great recently decide amazing husband smoke past close glad stop |
| 18 | Food | 0.05133 | eat ice cream chocolate make cake cookie breakfast favorite delicious buy bring bake chip treat dessert piece candy bowl bar |
| 56 | Family | 0.05124 | home day happy bring back make year feel great face return husband remember miss time moment memory parent smile stay |
| 63 | Other | 0.05110 | make laugh happy funny joke talk friend good learn daughter share lot thing coworker fight word people worker hear big |
| 53 | Exercise | 0.05090 | run bike ride walk mile hike long park yesterday rid beautiful mountain time nice trail morning great drive enjoy hiking |
| 10 | Entertain-ment | 0.05090 | movie watch friend enjoy good theater film night yesterday star girlfriend fun galaxy lot beauty favorite favourite love great cinema |
| 66 | Entertain-ment | 0.05086 | game win team play watch baseball match basketball favorite soccer league son final score tournament playoff cricket ball hit beat |
| 17 | Shopping | 0.05082 | buy phone computer problem laptop fix work purchase figure mobile issue finally internet solve today system iphone cell save smartphone |
| 76 | Exercise | 0.05070 | gym good feel great today workout hair work morning exercise personal yoga start cut record lift session class run haircut |
| 8 | Work | 0.05045 | work home day minute back drive run traffic light morning stop spot turn lot start decide time night walk commute |
| 75 | Food | 0.05038 | coffee drink morning beer cup good tea free wine friend shop buy favorite starbucks bar bottle enjoy local nice glass |
| 71 | Shopping | 0.05025 | receive mail order today arrive package wait amazon check free card gift expect send deliver online week yesterday letter email |
| 15 | Other | 0.04999 | love happy friend good people feel feeling life day boy moment wonderful girl meet give person make child amazing donate |
| 68 | Other | 0.04969 | make thing happy good lot feel people time love put life learn kind change small bad happen matter enjoy situation |
| 14 | Family | 0.04927 | moment happy life feel good time day family give brother person dream reach till proud unforgettable change goal true end |
| 65 | Gardening | 0.04764 | plant garden yard flower start tree air grow lawn rain water nice bloom smell spring mow seed finally grass bath |
| 7 | Entertain-ment | 0.04736 | song listen music favorite concert play band sing hear guitar dance album radio perform ticket release learn artist single podcast |
| 2 | Exercise | 0.04676 | lose week goal weight pound start month diet finally work step exercise find achieve reach day scale daily gain weigh |

**Table A1.** *Cont.*

| Topic | Label | Weight | Key Words |
|---|---|---|---|
| 77 | Other | 0.04645 | find start happy month give back plan free mother today happen feel decide completely lie laugh finally end call business |
| 37 | Family | 0.04593 | make happy mother sister father lunch law time son home wife roti part daughter morning serve brother leave learn cook |
| 51 | Entertainment | 0.04538 | book read finish write start find library note story learn year great local comic enjoy reading article paper publish chapter |
| 48 | Pets | 0.04511 | cat bed great fall cuddle night sit dog fun lay colleague snuggle asleep sleep discussion couch wake lap morning love |
| 78 | Vacation | 0.04494 | place family trip enjoy tour beautiful lot day travel city friend time visit hill zoo station uncle abroad animal experience |
| 32 | Religion | 0.04492 | temple family church easter enjoy yesterday god festival people sunday service member egg attend kid volunteer pray hunt trump morning |
| 61 | Education | 0.04434 | day happy school moment event enjoy class friend time remember pass life memorable exam lot finally college hear talk surprise |
| 23 | Other | 0.04432 | day wait man brother walk feel hand sit lady bus young road reach stand time line thing suddenly front house |
| 24 | Other | 0.04426 | happiness people give good happy feel great life world feeling person mind sense bring create joy share experience question satisfaction |
| 52 | Other | 0.04288 | nice day office meet parent care share smile today yesterday summer grandma trip cute life uncle send neighbor future ready |
| 1 | Shopping | 0.04277 | happy time feel purchase open great find house box hand hold parent excited buy finally home thing day man door |
| 39 | Party | 0.04168 | day life good dad give surprise happy present forget birthday moment surprisingly make date phone smart minute put end excited |
| 43 | Entertainment | 0.04078 | win ticket dollar lottery scratch competition money prize buy place contest casino free small give happy poker son tournament participate |
| 29 | Other | 0.03996 | happy time result feel parent brother great excited future month thing young expectation place mark back important anxiety idea small |
| 22 | Other | 0.03532 | life happiness people experience thing full part concept event live happy avoid show occasion tradition happen quality focus science seek |
| 69 | Other | 0.03194 | good work happiness time turn improve life virtue important research increase day flourish term live interest feedback sense skill advance |
| 40 | Other | 0.02854 | happiness happy state joy positive research mental include economic define pleasant share person fill big range reflect heart face intense |

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
