# Peer review of "Computing Happiness from Textual Data"

_stats, doi:10.3390/stats2030025_

Round 1
Reviewer 1 Report
This paper seems more an exercise for testing different very standard techniques and available software. There is no improvement over the literature to the best of my understanding. It would have been nice to also test the same models on different data sets. For example, doing some automatic crawling on Twitter through emoticons etc just to test the accordance at least on the happiness.
Overall the paper is well written and the analysis correct. It is a nice read for the beginners in the fields but I don't see any additional benefit.
Reviewer 2 Report
The paper analyzes a textual dataset to answer research questions about the causes of happiness with respect to differences of people regarding their sex, parenthood, marital status and age. They model each research question as a classification problem and explore the use of different learning features and classification models.
Though the results are interesting, especially with respect to the language and words used, they are somewhat expected. However, they can promote further research and potential applications and thus are useful.
The paper is generally well written with some minor mistakes. However, some parts need further explaining for the work to be self-contained and the reader to gain a better understanding of the methodology used.
--- The Introduction Section only states the goal of the research and does not describe the research methodology followed as it should.
--- In section 2.2, for topic modeling the authors use in the first approach three different metrics reported in bibliography. For the paper to be self-contained it would be better if the metrics were at least mentioned and briefly described and not just referenced.
--- In section 3.4.4., the concept of word embeddings should be explained.
No related work is analyzed or compared with the proposed methodology. While the treatment of the subject seems novel, the authors should include related work to put their contribution into a more clear context. For instance, related research on the HappyDB dataset such as those presented in the Affective Content Analysis workshop at AAAI2019, i.e.,
--- Rajendran et al, Happy together: Learning and understanding appraisal from natural language.
--- J. Wu et al., CruzAffect at AffCon 2019 Shared Task: A feature-rich approach to characterize happiness.
--- Syed et al., Ingredients for happiness: Modeling constructs via semi-supervised content driven inductive transfer learning.
In the paper of Asai et al., topics (more general categories) are defined for the given dataset. It would be interesting to see how these topics compare with the topics derived by LDA, and if the topics associated with male or female, parenthood and so on, are of the same or different more general categories so as to gain a higher level understanding.
Minor Comments
In page 5:
line 122: "ngrams" should be "bigrams"
lines 134-135: "The maximum likelihood method is commonly used to estimate these coefficients and obtain fit the model." should be rephrased.
line 137: "we by basically plug-in..." should be "we basically plug-in.."
In page 6,
lines: 161 to 164. This should be rephrased, either as steps of an algorithm or with the use of correct syntax... i.e., "Given the current model, we compute...."
Reviewer 3 Report
It's interesting to explore the categories of the happy moment and the relationship of the happy moment with the social features. The data and methods are clearly explained in the paper and it's creative to use dependency to extract more meaningful bigrams.
It’s not clear why the score thresholds are 1/3 and 3 for common or not. For example, the score in Table 6 varies from 11875 to 0.036. Why would 11875 to 3 belong to one class and 0.3333 to 0.036 belongs to the other?
Why country feature is not discussed?
Overall, it's interesting to read the different lexical information for different social features.
Reviewer 4 Report
Overall, the authors present very thorough analyses around happiness and finding patterns that are different across gender, marital status, and parenthood.
My main substantive comments revolve around clarifying the methods utilized in this paper. For instance, the authors cite references in certain parts that manuscript that benefit from further describing critical details on methods.
For example, on page 2, the authors state that Amazon Turk workers were asked the following question: “What made you happy in the past 24 hours (or alternatively, the past 3 months)? (Asai et al. [2])." The two time points are confusing. How many people were asked about the past 24 hours and how many people were asked about the past 3 months? How was the differing time frame handled in the analysis? Also, On manuscript page 4 and 5, the authors reference metrics provided by the following articles (Griffiths2004, CaoJuan2009, Arun2010), but don't describe them. This is confusing because these metrics are then used to select the number of topics, so it would be important to describe these metrics.
Table 3 has some unexpected words like “tirupati” and “stardew.” These words do not seem very common. Could the authors explain these findings?
In Table 9, what is meant by “3mth?”
The manuscript ends a bit abruptly. Perhaps the discussion can be extended to further describe study limitations and study implications.
Minor comment: Line 330, the word “response” is misspelled
